# Alterations in Patients’ Clinical Outcomes and Respiratory Viral Pathogen Activity following the COVID-19 Pandemic

**DOI:** 10.3390/v15101975

**Published:** 2023-09-22

**Authors:** Khaled Al Oweidat, Ahmad A. Toubasi, Ahmad Alghrabli, Yasmeen Khater, Noor Saleh, Asma S. Albtoosh, Rawan Shafeek Batarseh

**Affiliations:** 1Department of Respiratory and Sleep Medicine, Department of Internal Medicine, School of Medicine, The University of Jordan, Amman 11942, Jordan; k.oweidat@ju.edu.jo (K.A.O.); nourhanisaleh@hotmail.com (N.S.); asmaalbtoosh@gmail.com (A.S.A.); 2Faculty of Medicine, The University of Jordan, Amman 11942, Jordan; 3Department of Internal Medicine, School of Medicine, The University of Jordan, Amman 11942, Jordan; ahmmm197@hotmail.com (A.A.); yasmeenkhater96@gmail.com (Y.K.); 4Department of Microbiology, Jordan University Hospital, Amman 11942, Jordan; r.batarseh@ju.edu.jo

**Keywords:** human, COVID-19, pandemic, viruses

## Abstract

Background: Before the COVID-19 pandemic, respiratory pathogens such as influenza, parainfluenza, and respiratory syncytial virus were the most commonly detected viruses among hospitalized patients with respiratory tract infections. Methods: This was a retrospective observational study of inpatients and outpatients who attended Jordan University Hospital and underwent Nasopharyngeal Aspiration (NPA) in the periods from December 2017 to December 2018 and from December 2021 to December 2022. The results of multiplex respiratory pathogen real-time PCR tests for nasopharyngeal swab specimens were extracted from the electronic-based molecular diagnostic laboratory record of JUH. We compared the prevalence of the detected viruses as well as the patients’ characteristics and outcomes between the two periods. Results: The total number of included patients was 695. Our analysis showed that a higher percentage of patients with hypertension and diabetes presented before the pandemic compared to the same period after it (*p*-value < 0.001). The need for O2 devices, white blood cell counts, diastolic blood pressure, and the length of hospital stay were significantly higher among patients who presented before the pandemic (*p*-value < 0.050). Influenza H1N1 (8.70% vs. 4.03%), influenza B (1.67% vs. 0.25%), parainfluenza (1.00% vs. 0.00%), human metapneumovirus (5.35% vs. 0.76%), adenoviruses (6.35% vs. 3.02%), and coronaviruses (8.70% vs. 3.53%) were detected with higher frequency in the period before the pandemic (*p*-value = 0.011, 0.045, 0.045, 0.000, 0.035, 0.004). These results were similar in terms of changes in the detection rates of viruses after matching the number of tested patients between the periods before and after the pandemic. Conclusions: We have demonstrated a reduction in the detection of several viruses, which might be due to the increase in public awareness toward infection protection measures after the COVID-19 pandemic.

## 1. Introduction

Since the start of the pandemic, the 2019 coronavirus disease (COVID-19) has emerged as one of the most severe infectious disease pandemics in recent history [1]. Over the course of several years, it has inflicted hundreds of millions of cases and led to millions of deaths worldwide [2]. The rapid spread of the virus has caused a decline in life expectancy in countries like the United States of America (USA) [3]. leaving a lasting impact on global health and societies.

The emergence of COVID-19 in late 2019 as a novel coronavirus created an unprecedented public health challenge for the world. The virus, Severe Acute Respiratory Syndrome Coronavirus 2 (SARS-CoV-2), quickly spread from its origins in Wuhan, China, to virtually every corner of the globe. In the face of this unprecedented threat, countries implemented various measures, such as lockdowns, travel restrictions, mask mandates, and social distancing guidelines, in an attempt to mitigate the spread of the virus and reduce the burden on healthcare systems.

The swift and widespread implementation of these measures was an extraordinary public health effort, aiming to curb the transmission of COVID-19 and protect vulnerable populations. Also, the medical efforts made to provide care for COVID-19 patients at the beginning of the pandemic burdened resources [4]. Countries adopted a wide range of strategies tailored to their specific circumstances, leading to a global reduction in mobility and social interactions. While these interventions were primarily directed at controlling COVID-19, they also inadvertently impacted the transmission of other respiratory viruses.

Before the COVID-19 pandemic, respiratory pathogens such as influenza, parainfluenza, respiratory syncytial virus (RSV), human metapneumovirus (hMPV), and human coronavirus were among the common viruses detected in patients with respiratory tract infections [5,6,7]. The seasonal occurrence of these viruses has been well established, with yearly fluctuations in their prevalence during specific months or seasons. However, with the sudden emergence of COVID-19 and its rapid global transmission, the dynamics of respiratory virus epidemiology were expected to be affected.

The COVID-19 pandemic prompted governments and health authorities worldwide to take unprecedented actions to contain the spread of the virus. Lockdowns, travel restrictions, school closures, and the widespread use of face masks became standard strategies to control the outbreak. These measures aimed to reduce person-to-person transmission and limit the strain on healthcare facilities. Consequently, it was anticipated that the prevalence of other respiratory viruses would decline due to reduced contact and increased adherence to hygiene practices [8].

Several studies have investigated the impact of COVID-19 preventive measures on the prevalence of seasonal respiratory viruses. The results have been encouraging, with a substantial reduction observed in the prevalence of these viruses during the COVID-19 pandemic [9,10,11]. This decline in cases has been attributed to the measures taken to combat COVID-19, indicating that such strategies may have broader implications for controlling other respiratory infections in the future.

The increase in public health knowledge about COVID-19 and the adoption of protective measures have been notable observations during the pandemic [12,13,14]. People of all age groups, from the elderly to school and university students, demonstrated heightened awareness and compliance with preventive behaviors. The rapid dissemination of accurate information through various channels, including the media, public health campaigns, and online resources, contributed to this widespread awareness. Additionally, studies have shown that there was a high intention to continue adopting protective behaviors even after the pandemic [15]. This collective shift in behavior has led to the concept of the “new normal”, where certain pandemic-induced changes become ingrained in daily life, shaping long-term behavioral patterns.

As the pandemic unfolded, the world witnessed changes in various aspects of life. Telecommuting became more prevalent, air travel declined, and a heightened focus on personal hygiene and public health emerged. The concept of the “new normal” described this transformation, indicating that post-pandemic life might be significantly different from pre-pandemic times. As the world adapted to this new reality, it raised questions about how these changes might impact the future epidemiology of respiratory viruses [15].

Given the increased awareness and adoption of new behaviors during the pandemic, it is reasonable to hypothesize that these changes in population thoughts and behaviors may impact the epidemiology of seasonal respiratory viruses. While numerous studies have investigated the changes in respiratory virus prevalence during the pandemic, to the best of our knowledge, none have examined the subsequent epidemiological shifts after the pandemic. Hence, this study aims to investigate the alterations in the viruses detected in patients presenting with respiratory infection symptoms before and after the COVID-19 pandemic. Additionally, we seek to explore potential differences in patients’ demographics, laboratory investigations, and outcomes between these two distinct periods.

## 2. Methods

This retrospective observational study focuses on inpatients and outpatients who sought medical attention at Jordan University Hospital (JUH) Emergency Department (ED), a tertiary adult and pediatric hospital located in Amman, Jordan. Patients from all age groups who underwent nasopharyngeal aspiration (NPA) with viral polymerase chain reaction (PCR) were included in the study, covering the periods from 1 May 2017 to 30 June 2018 and from 1 May 2022 to 30 June 2023. NPA with PCR for pathogen detection was performed on patients who presented on their own (not transferred from another hospital) with symptoms of respiratory viral infection (febrile respiratory illness), based on the ED physician’s decision. Patients were excluded from the study if they had a COVID-19 infection (tested by PCR nasal swab) or immunosuppression, including transplant patients, patients with cancer, and those receiving immunosuppressive drugs or chemotherapy. The primary aim of the study was to investigate the differences in the viruses detected from patients’ samples before and after the COVID-19 pandemic, while the secondary aims were to explore differences in patients’ demographics, laboratory investigations, and outcomes between the two periods. To minimize the impact of seasonality, an equivalent number of months were selected before and after the COVID-19 pandemic.

The chosen period before the pandemic extended from 1 May 2017 to 30 June 2018, while the period after it was from 1 May 2022 to 30 June 2023. These specific periods were selected as they represent the furthest time points from the peaks of the COVID-19 pandemic, thereby reducing the effect of COVID-19 spread on the distribution of detected viruses.

This study adhered to human ethical practices and the principles outlined in the Declaration of Helsinki. The Institutional Review Board (IRB#223000353) at the University of Jordan approved the study, waiving the need for informed consent.

## 3. Study Protocol

Patient data, including demographics, comorbidities (e.g., hypertension, diabetes, respiratory diseases, and cardiovascular conditions), COVID-19 infection history, COVID-19 vaccination status, and the type of COVID-19 vaccine received, were collected from medical records. Vital signs on presentation, such as systolic and diastolic blood pressure, heart rate, respiratory rate, temperature, and O2 saturation, were also recorded.

## 4. Follow-Up and Patients’ Outcomes

Laboratory investigations, including C-reactive protein levels and white blood cell (WBC) counts, as well as neutrophil and lymphocyte percentages, were obtained from patient records. Information regarding patients’ need for O2 devices during hospital stays, rates of intensive care unit (ICU) admission, length of hospital stay, and final outcomes was also collected.

## 5. Virus Detection

The same PCR tests were used in both periods with the same frequency of testing. The results of multiplex respiratory pathogen real-time PCR tests for nasopharyngeal swab specimens were extracted from the electronic-based molecular diagnostic laboratory record of JUH. Nasopharyngeal swabs were collected using the universal transport and preservation Copan media and stored at −20 °C. Conventional real-time PCR, specifically the FTD Respiratory Pathogens 21 Assay Kit and EZ 1 and 2 Virus Mini Kit V2.0 by Qiagen, was employed for nucleic acid extraction. The viruses detected in this study included human adenovirus, human rhinovirus and enterovirus, human parainfluenza viruses 1–4, respiratory syncytial virus (RSV), metapneumovirus A and B (MNPV), influenza A, A/H1N1, and B, as well as coronaviruses 229E, NL63, OC43, and HKU1.

## 6. Data Analysis

Patient data were recorded in Microsoft Office Excel 2019 and later imported into IBM SPSS v.25 software for analysis. Continuous variables were summarized as mean and standard deviation, while categorical variables were summarized as counts and percentages. To assess differences between the two periods, chi-square tests and T-tests were conducted as appropriate. In addition, we carried out a random matching process between the two periods to equalize the number of tests between the two groups. A random sample that equals the before-pandemic group in number was taken from the after-pandemic group. This was followed by assessing the differences between the two groups in the virus distribution using chi-square. A *p*-value less than 0.05 was considered statistically significant for all tests.

## 7. Results

### 7.1. Characteristics of the Included Patients

The total number of included patients was 695; 54.4% of them were males among pediatric patients, while 48.0% were males among adults. The prevalence of hypertension and diabetes was 11.2% and 41.5% among adults, whereas it was 1.1% and 2.0% among pediatrics, respectively. The percentages of respiratory and cardiovascular diseases were 30.9% and 29.3% among adults and 25.3% and 5.6% among pediatric patients, respectively. The mean systolic and diastolic blood pressures were 122.69 ± 21.41 mmHg and 72.33 ± 13.07 mmHg among adults, while the means were 142.73 ± 81.7 mmHg and 61.54 ± 10.89 among pediatric patients, respectively. Among adults, the mean body temperature was 37.12 ± 0.82c and the mean O2 saturation was 91.45 ± 7.61%. The mean body temperature and O2 saturation among pediatric patients were 36.98 ± 3.05 and 92.63 ± 6.04, respectively. Table 1 shows the characteristics of the included patients, while Table 2 demonstrates the characteristics of the COVID-19 infection and vaccination history.

### 7.2. Laboratory Investigations and Patients’ Outcomes

The majority of the patients were hospitalized (98.4%), and 23.0% of the patients were admitted to the ICU among adults. Similarly, 99.6% were hospitalized and 36.1% were admitted to the ICU among pediatric patients. The prevalence of hypertension and diabetes among adults admitted to the ICU was 47.4% and 35.1%, respectively, while it was 3.1% and 5.6% among pediatric patients. Moreover, 60.9% of the adult patients needed O2 devices, and 14.2% of them needed intubation. Among adults, the mean CRP was 85.76 ± 97.75 and the mean WBC count was 11.55 ± 9.58, while among pediatric patients, the mean CRP and WBC were 33.79 ± 54.85 and 12.47 ± 9.79, respectively. In addition, the mean length of the hospital stay was 11.45 ± 11.02 among adults, while it was 12.47 ± 9.79. Moreover, 13.8% and 3.4% died among adult and pediatric patients, respectively. Table 3 displays the laboratory investigation results, patients’ outcomes, and characteristics of patients admitted to the ICU.

### 7.3. Differences in the Characteristics and Outcomes of the Included Patients between the Periods before and after the Pandemic

Our analysis showed that a higher percentage of patients with hypertension presented before the pandemic (24.6%) compared to the same period after it (12.6%) (*p*-value < 0.001). Also, there was a significant difference in the percentage of patients with diabetes between the period before and after the pandemic (*p*-value = 0.001), as a significantly higher percentage of patients with diabetes presented before the pandemic (21.3%) compared to after it (12.1%). The need for O2 devices was significantly higher among patients who presented before the pandemic (65.1%) compared to after it (51.1%). Patients who presented before the pandemic had significantly higher diastolic blood pressure at presentation (*p*-value < 0.001). On the other hand, patients presented before the pandemic had significantly higher WBC counts compared to patients presented after the pandemic (*p*-value = 0.008). The mean length of hospital stay was significantly higher among patients presented before the pandemic (12.21 ± 20.24 days) compared to after it (9.47 ± 12.26 days) (*p*-value = 0.038) (Table 4).

### 7.4. Differences in the Detected Viruses between the Periods before and after the Pandemic

The most frequently detected viruses in patients with NPA were rhinoviruses/enteroviruses (20.69%), followed by RSV (16.81%), influenza A (9.9%), influenza H1N1 (6.03%), and coronavirus (5.75%) (Figure 1). There were significant differences in the distribution of the detected viruses between the periods before and after the pandemic. Influenza H1N1 was detected significantly higher in the period before the pandemic (8.70%) compared to the period after it (4.03%) (*p*-value = 0.011). Moreover, influenza B and coronaviruses were detected at a higher frequency in the period before the pandemic compared to after it (1.67% vs. 0.25%, 8.70% vs. 3.53%) (*p*-value = 0.045, 0.004). Parainfluenza was also detected in a higher percentage in the period before the pandemic (1.00%) than after it (0.00%) (*p*-value = 0.045). Furthermore, HMPV and adenoviruses were detected more frequently in the period before the pandemic than after it (5.35% vs. 0.76%, 6.35% vs. 3.02%) (*p*-value < 0.001, 0.035). After matching the number of tested patients before and after the pandemic, similar results to the primary analysis were found (Table 4). The differences in the detected viruses between the periods before the pandemic and after it are demonstrated in Figure 2.

## 8. Discussion

The end of the COVID-19 pandemic presents an unprecedented opportunity to assess the impact of public health interventions, vaccination, protective measures awareness campaigns, and changes in human social activities on the epidemiologic characteristics of other circulating respiratory viruses. In addition, it is important to assess the differences in the characteristics and outcomes of the patients infected by these viruses.

There was a higher percentage of patients with diabetes and hypertension who presented before the pandemic than after it. These findings can be explained by the increase in knowledge after the pandemic that the elderly population, which has a higher prevalence of hypertension and diabetes, is more susceptible to severe viral infections [16]. As a result, higher compliance with infection-protective measures such as face masks and avoidance of social gatherings among the elderly after the pandemic has been found in observational studies [17,18]. Moreover, we found a significant reduction in the length of hospital stay and the need for O2 devices for hospitalized patients in the period after the pandemic, suggesting that viral infections might be milder. Previous studies suggested that influenza seasons before the pandemic were severer and associated with a higher rate of hospitalization [8,19]. Another finding in our study that might support this evidence is that there was a lower mean diastolic blood pressure and white blood cell counts among patients in the period after the pandemic [20,21].

Our results demonstrated that the detection of influenza H1N1, influenza B, coronaviruses, parainfluenza, HMPV, and adenoviruses was significantly lower in the period after the pandemic. Previous studies conducted in the USA and Japan showed similar results during the COVID-19 pandemic [8,19]. School closures, suspension of large events, and awareness of measures that reduce viral infection transmission shifted public behaviors [17], as studies showed that public behaviors have changed significantly due to the COVID-19 pandemic [17,18,22]. Public populations started to avoid public transportation and wash and sanitize their hands more after the pandemic compared to before it [23]. In addition, studies found that people avoided gatherings when they were having respiratory infection symptoms and wore masks more in the period after the pandemic [23]. Furthermore, the shift in meetings and educational activities to virtual platforms instead of face-to-face during the pandemic familiarized the populations with the availability of these technological advancements [24]. Due to its advantages, including comfort and accessibility, it was expected that this shift might continue even after the pandemic [25]. Studies show that learners are ready and willing to make greater use of online educational platforms and virtual meetings [26]. Face-to-face educational activities early in the COVID-19 pandemic have been implicated in the huge increase in infection rates [27]. Another important factor is the uptake of the COVID-19 vaccination, which increased with time as a result of increased awareness [28]. The prevalence of COVID-19 vaccination in our study was high, reaching 90.7%, which is similar to the rates reported in the WHO dashboard [1]. Moreover, studies that compared influenza virus activity years before the COVID-19 pandemic and during it showed that influenza activity has declined significantly, suggesting that measures taken during the pandemic were effective in reducing the spread of other viral respiratory diseases [29]. Similar results were found in Korea, Taiwan, and Singapore [27,29,30]. Additionally, studies also showed that not only the incidence of influenza viruses decreased but also their severity and infectivity [19,27,30]. Moreover, a study conducted in Korea demonstrated that not only influenza virus incidence has declined but also another seven respiratory viruses, including adenoviruses, respiratory syncytial virus, HMPV, rhinoviruses, parainfluenza viruses, and coronaviruses [29]. It is important to highlight that we found that most of the influenza viruses were caused by influenza A, especially the H1N1 strain, and only a minority of the infections were caused by influenza B, a pattern that was noticed in the period before and after the pandemic. This is similar to the findings in the study carried out in Korea [29]. However, in a study conducted in the USA, the majority of the circulating influenza viruses were caused by influenza B [31]. Also, in a study conducted in the UK, influenza A H2N3 dominated during the pandemic period [32]. In addition, hospitalization rates were lower during the pandemic compared to previous seasons, which strengthens the evidence of a decreased virulence of the infection [27,33]. On the other hand, it is important to highlight that these reductions in virus incidence might not only be due to the improvement of public awareness and changes in human behaviors. Ecological studies revealed that climate change in the past 40 years has fastened significantly [34]. An increase in temperature across the globe has been implicated in decreasing the infectivity and virulence of several viruses, including influenza viruses [35].

This is one of the first studies to assess the impact of the COVID-19 pandemic on the change in the viruses detected in respiratory infection patients by comparing the detection rates between the periods before and after the pandemic. Few studies were conducted to investigate the change in the virus epidemiology between the periods before and during the COVID-19 pandemic [36,37,38]. The majority of these studies were conducted in Europe, China, and America, with very few studies conducted in the Middle East region [36,37,38]. Our study adds to the literature by being conducted in a region with a scarcity of data. In addition, it is one of the few studies that compared the changes in virus epidemiology between the periods before and after the COVID-19 pandemic and not between the periods before and during the pandemic. However, several limitations should be acknowledged. First, the retrospective design of our study limits our results to associations and not causation. Second, this study was conducted at a single center, which limits the generalizability of our findings. Moreover, this study was conducted in a tertiary referral hospital, while the PCR test is considered high-cost and low-availability. Thus, our results might be shifted to patients with more severe infections, which is indicated by the high hospitalization rate. Yet, it is important to note that these limitations were in the before- and after-pandemic periods; hence, they are less likely to affect our results. Third, the reduction in the detection of viruses after the pandemic might be a result of reduced testing due to burnout among healthcare providers and the high demand for resources during the pandemic period. However, the definition of respiratory viral infection disease is based on clinical symptoms, and physicians are expected to carry out the testing on any patient with similar symptoms. Finally, the kits used in our study do not include all the viruses and their strains, which might affect our results.

In conclusion, we showed that detection of influenza H1N1, influenza B, coronaviruses, parainfluenza, HMPV, and adenoviruses was significantly lower in the period after the pandemic compared to before it. In addition, we found that after the pandemic, a lower percentage of patients with diabetes and hypertension presented with viral respiratory infections, as well as a reduction in the length of hospital stay and the need for O2 devices among hospitalized patients. The increase in public awareness toward infection protection measures during the COVID-19 pandemic and the changes in human behaviors may have resulted in a reduction in the rates of detecting several viruses.

## Figures and Tables

**Figure 1 viruses-15-01975-f001:**
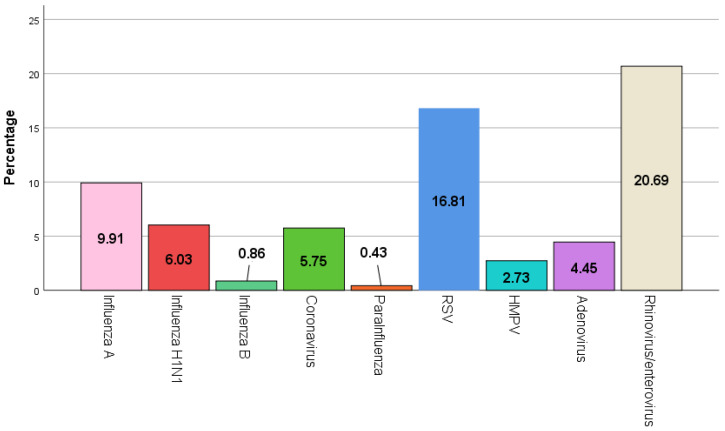
Distribution of viruses in patient samples. Influenza A: influenza type A other than H1N1; Influenza H1N1: influenza subtype hemagglutinin1 neuraminidase 1; Influenza B: influenza type B; RSV: respiratory syncytial virus; HMPV: human metapneumovirus.

**Figure 2 viruses-15-01975-f002:**
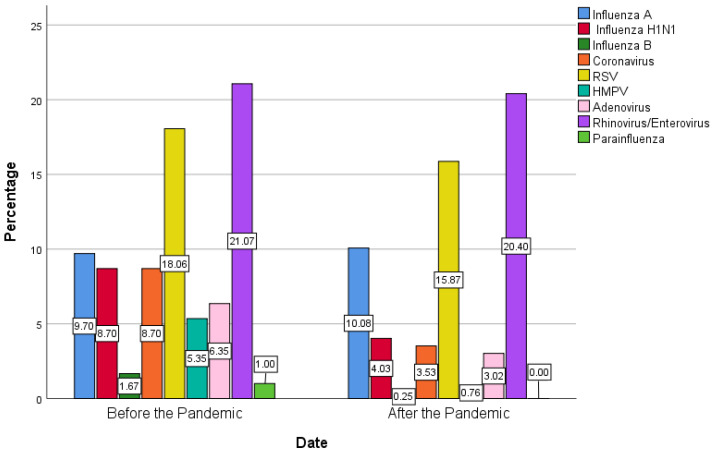
Distribution of detected viruses between the periods before and after the pandemic. Influenza A: Influenza type A other than H1N1; Influenza H1N1: influenza subtype hemagglutinin1 neuraminidase 1; Influenza B: influenza type B; RSV: respiratory syncytial virus; HMPV: human metapneumovirus.

**Table 1 viruses-15-01975-t001:** Ggeneral demographics of the participants.

Variable	Response		Pediatric Patients		Adults	
			Frequency	Percentage (%)	Frequency	Percentage (%)
Sex	Male		243	54.4	119	48.0
	Female		204	45.6	129	52.0
Hypertension	No		442	98.9	221	88.8
	Yes		5	1.1	28	11.2
Diabetes Mellitus	No		437	98.0	144	58.5
	Yes		9	2.0	102	41.5
Respiratory Diseases	No		334	74.7	172	69.1
	Yes		113	25.3	77	30.9
Cardiovascular Diseases	No		422	94.4	176	70.7
	Yes		25	5.6	73	29.3
Variable	Pediatric Patients		Adults			
	Mean	SD	Mean	SD		
Systolic Blood Pressure (normal for pediatric patients: 60–131, for adults: 90–120)	142.73	81.7	122.69	21.41		
Diastolic Blood Pressure (normal for pediatric patients: 31–83, for adults: 60–80)	61.54	10.89	72.33	13.07		
Heart Rate (normal for pediatric patients: 60–200, for adults: 60–100)	130	56.85	95.77	20.52		
Respiratory Rate (normal for pediatric patients: 12–60, for adults: 12–18)	31.11	11.99	21.86	5.75		
Temperature (normal: 36.5–37.3c)	36.98	3.05	37.12	0.82		
O2 Saturation (normal: 95–100%)	92.63	6.04	91.45	7.61		

**Table 2 viruses-15-01975-t002:** COVID-19 infection and vaccination history among patients in the after-pandemic group.

Variable	Response	Frequency	Percentage (%)
COVID-19 History	No	340	85.6
	Yes	57	14.4
COVID-19 Vaccination History	No	37	9.3
	Yes	358	90.7
COVID-19 Vaccine Types	Sinopharm	150	37.8
	Astrazeneca	79	19.9
	Pfizer	168	42.3

**Table 3 viruses-15-01975-t003:** Patient outcomes and laboratory investigations.

Variable	Response		Pediatric Patients		Adults	
			Frequency	Percentage (%)	Frequency	Percentage (%)
Hospitalization	No		2	0.4	4	1.6
	Yes		445	99.6	245	98.4
ICU Admission	No		285	63.9	191	77.0
	Yes		161	36.1	57	23.0
Characteristics of the Patients Admitted to the ICU	Hypertension		5	3.1	27	47.4
	Diabetes		9	5.6	20	35.1
	Cardiovascular Diseases		25	15.5	29	50.9
	Respiratory Diseases		27	16.7	27	47.4
Need of O2 Devices	No		201	45.0	97	39.1
	Yes		246	55.0	151	60.9
Type of O2 Device	Face Mask		21	4.7	41	16.5
	High Flow Nasal Canula		44	9.9	20	8.0
	BiPAP		40	9.0	54	21.7
	CPAP		90	20.2	25	10.0
	Intubation		44	9.9	35	14.2
Death	No		432	96.6	213	86.2
	Yes		15	3.4	34	13.8
Variable	Pediatric Patients		Adults			
	Mean	SD	Mean		SD	
Length of Hospital Stay (days)	10.20	18.49	11.45		11.02	
C-reactive Protein (mg/dL) (normal: <0.3 mg/dL)	33.79	54.85	85.76		97.75	
White Blood Cells (10^3^) (normal: 4.5 × 10^3^ to 11.0 × 10^3^)	12.47	9.79	9.92		8.98	
Neutrophils (normal: 40–60%)	50.73%	21.83%	71.05%		16.76%	
Lymphocyte (normal: 20–40%)	39.07%	20.03%	19.49%		14.20%	

**Table 4 viruses-15-01975-t004:** Differences in the demographics between patients before and after the pandemic. A *T*-test and a Chi-square test were used to investigate the differences.

Variable		Before Pandemic (n = 298)	After Pandemic (n = 397)	*p*-Value
Gender	Male	147 (49.3)	215 (54.2)	0.207
	Female	151 (50.7)	182 (45.8)	
Age		55.13 ± 21.52	59.08 ± 19.06	0.353
Respiratory Diseases	No	220 (73.6)	286 (72.0)	0.652
	Yes	79 (26.4)	111 (28.0)	
Hypertension	No	224 (75.4)	346 (87.4)	0.000 *
	Yes	73 (24.6)	50 (12.6)	
Diabetes	No	233 (78.7)	348 (87.9)	0.001 *
	Yes	63 (21.3)	48 (12.1)	
Cardiovascular Diseases	No	253 (84.6)	345 (86.9)	0.391
	Yes	46 (46.9)	52 (53.1)	
Test Results	No	107 (35.8)	165 (41.6)	0.122
	Yes	192 (64.2)	232 (58.4)	
Coinfection	No	250 (83.6)	332 (83.7)	0.900
	Yes	49 (16.4)	65 (16.3)	
Viruses (Before matching the sample size of the two groups)	Influenza A	29 (9.7)	40 (10.1)	0.869
	Influenza H1N1	26 (8.7)	16 (4.0)	0.011 *
	Influenza B	5 (1.7)	1 (0.3)	0.045 *
	Coronavirus	26 (8.7)	14 (3.5)	0.004 *
	Parainfluenza	3 (1.0)	0 (0.0)	0.045 *
	RSV	54 (18.1)	63 (15.9)	0.444
	HMPV	16 (5.2)	3 (0.8)	0.000 *
	Adenovirus	19 (6.4)	12 (3.0)	0.035 *
	Rhinovirus/Enterovirus	63 (21.1)	81 (20.4)	0.830
Viruses (after matching the sample size of the two groups)	Influenza A	29 (9.7)	26 (8.7)	0.681
	Influenza H1N1	26 (8.7)	10 (2.98)	0.048 *
	Influenza B	5 (1.7)	1 (0.3)	0.045 *
	Coronavirus	26 (8.7)	6 (2.0)	0.000 *
	Parainfluenza	3 (1.0)	0 (0.0)	0.045 *
	RSV	54 (18.1)	55 (50.5)	0.900
	HMPV	16 (5.2)	2 (0.7)	0.001 *
	Adenovirus	19 (6.4)	7 (2.7)	0.049
	Rhinovirus/Enterovirus	63 (21.1)	63 (21.1)	1.000
Hospitalization	No	1 (0.3)	5 (1.3)	0.191
	Yes	298 (99.7)	392 (98.7)	
ICU Admission	No	193 (65.0)	283 (71.3)	0.077
	Yes	104 (35.0)	114 (28.7)	
Use of O2 Devices	No	104 (34.9)	194 (48.9)	0.000 *
	Yes	194 (65.1)	203 (51.1)	
Intubation	No	266 (89.0)	348 (88.3)	0.793
	Yes	33 (11.0)	46 (11.7)	
Death	No	276 (92.6)	369 (93.2)	0.774
	Yes	22 (7.4)	27 (6.8)	
Systolic Blood Pressure (normal for pediatric patients: 60–131, for adults: 90–120)		130.72 ± 30.59	115.88 ± 24.41	0.252
Diastolic Blood Pressure (normal for pediatric patients: 31–83, for adults: 60–80)		69.08 ± 12.69	62.65 ± 12.20	0.000 *
Heart Rate (normal for pediatric patients: 60–200, for adults: 60–100)		102.20 ± 29.37	107 ± 75.27	0.385
Temperature (normal: 36.5–37.3c)		37.13 ± 2.26	36.96 ± 2.66	0.371
O2 Saturation (normal: 95–100%)		92.07 ± 6.06	92.31 ± 7.08	0.620
Respiratory Rate (normal for pediatric patients: 12–60, for adults: 12–18)		23.35 ± 1.87	23.07 ± 1.94	0.320
C-Reactive Protein (normal: <0.3 mg/dL)		54.85 ± 84.04	50.64 ± 71.76	0.484
White Blood Cells (normal: 4.5 × 10^3^ to 11.0 × 10^3^)		10.49 ± 8.16	12.37 ± 10.48	0.008 *
Neutrophils (normal: 40–60%)		57.73 ± 22.77	58.20 ± 22.11	0.787
Lymphocytes (normal: 20–40%)		31.68 ± 20.70	32.35 ± 20.25	0.670
Length of Hospital Stay		12.21 ± 20.24	9.47 ± 12.26	0.038 *

* *p*-value < 0.050.

## Data Availability

The data associated with this manuscript are available from the corresponding author upon reasonable request.

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
