# Peer review of "Alterations in Patients’ Clinical Outcomes and Respiratory Viral Pathogen Activity following the COVID-19 Pandemic"

_viruses, 2023, doi:10.3390/v15101975_

Round 1

Reviewer 1 Report

Revision manuscript entitled “The Change in the Patients Clinical Outcomes and Respiratory Viral Pathogens Activity After the COVID-19 Pandemic”

In this manuscript, the authors report the prevalence of several respiratory viral pathogens in two population groups before and after the COVID-19 pandemic. In these groups, some laboratory parameters as well as comorbidities were also investigated. The results showed that the prevalence of the investigated respiratory viruses decreased after the pandemic probably due to the observance of preventive measures.

Patients’ population:

It is surprising that half of the patients who have a mean age of 22 years suffers from hypertension. Is that a fragile population? In addition, the prevalence of diabetes mellitus is quite high.

In Table 1, the total number of patients evaluated for each of the pathologies examined should be 695. Looking at each single line of the table the sum is not 695.

The COVID-19 history should be reported in another table containing only patients examined after the pandemic since the first group never encounter SARS-COV-2. Then, indicate how many among them were vaccinated and the specific vaccine.

Paragraph 7.2:

Indicate the age and the underlying medical conditions of the patients admitted to the ICU. In Table 2, check the numbers as in Table 1.

Discussion:

Second paragraph, delete “than” before pandemic.

Check the text for typos. Englisn language can be improved.

Author Response

Reviewer 1:

Revision manuscript entitled “The Change in the Patients Clinical Outcomes and Respiratory Viral Pathogens Activity After the COVID-19 Pandemic”

In this manuscript, the authors report the prevalence of several respiratory viral pathogens in two population groups before and after the COVID-19 pandemic. In these groups, some laboratory parameters as well as comorbidities were also investigated. The results showed that the prevalence of the investigated respiratory viruses decreased after the pandemic probably due to the observance of preventive measures.

- Response: Thank you so much for your efforts and time in reviewing the manuscript! We really appreciate it! We complied with the comments mentioned below.

Patients’ population:

It is surprising that half of the patients who have a mean age of 22 years suffers from hypertension. Is that a fragile population? In addition, the prevalence of diabetes mellitus is quite high.

- Response: Thank you so much for drawing our attention to this issue. We totally agree with you. There was a typo in the prevalence of hypertension and diabetes. We fixed this typo and the prevalence of both was 17.7%. We edited the results section and Table 1 accordingly;

“Results:

 The prevalence of hypertension and diabetes was 17.7% for each.”

In Table 1, the total number of patients evaluated for each of the pathologies examined should be 695. Looking at each single line of the table the sum is not 695.

- Response: Thank you so much for your comment. The difference from the total number of patients are the patients with missing data.

The COVID-19 history should be reported in another table containing only patients examined after the pandemic since the first group never encounter SARS-COV-2. Then, indicate how many among them were vaccinated and the specific vaccine.

- Response: Thank you for your suggestion. We totally agree with you. We reported these results in another table;

Table 2. COVID-19 Infection and Vaccination History among Patients in the After Pandemic Group

Variable

Response

Frequency

Percentage (%)

COVID-19 History

No

340

85.6

Yes

57

14.4

COVID-19 Vaccination History

No

37

9.3

Yes

358

90.7

COVID-19 Vaccine Types

Sinopharm

150

37.8

Astrazeneca

79

19.9

Pfizer

168

42.3

Paragraph 7.2:

Indicate the age and the underlying medical conditions of the patients admitted to the ICU. In Table 2, check the numbers as in Table 1.

- Response: Thank you so much for your suggestion. We added the characteristics of the patients admitted to the ICU on Table 3 and we checked the numbers in Table 1;

“Results:

The majority of the patients were hospitalized (99.1%) and 31.4% of the patients were admitted to the ICU. The mean age of the patients admitted to the ICU was 18.00 ± 28.51. The prevalence of hypertension and diabetes were 14.7% and 13.4%, respectively.”

“Table 3 displays the laboratory investigation results, patients’ outcomes and characteristics of patients admitted to the ICU.”

Table 3. Patients Outcomes and Laboratory Investigations.

Variable

Response

Frequency

Percentage (%)

Hospitalization

No

6

0.9

Yes

690

99.1

ICU Admission

No

476

68.6

Yes

218

31.4

Characteristics of the Patients Admitted to the ICU

Age

18.00 ± 28.51

Hypertension

32

14.7

Diabetes

29

13.4

Cardiovascular Diseases

38

17.4

Respiratory Diseases

54

24.8

Need of O2 Devices

No

397

57.1

Yes

298

42.9

Type of O2 Device

Face Mask

62

8.9

High Flow Nasal Canula

64

9.2

BiPAP

94

13.5

CPAP

115

16.5

Intubation

79

11.4

Death

No

645

92.9

Yes

49

7.1

Variable

Mean

SD

Range

Length of Hospital Stay

10.65

16.22

0-180

C-reactive Protein (mg/dl)

52.46

77.28

0.01-512

White Blood Cells (10^3)

11.55

9.58

1-133.0

Neutrophils

57.99%

22.38%

2.00%-98.00%

Lymphocyte

32.06%

20.44%

0.40%-99.00%

Discussion:

Second paragraph, delete “than” before pandemic.

- Response: Thank you for your comment! We edited the mentioned typo;

“Discussion:

There was a higher percentage of patients with diabetes and hypertension presented before the pandemic than after it.”

Reviewer 2 Report

-          The title should be revised for clearer English phrasing.

-          The term “isolated” is incorrectly used in relation to the diagnosis of viral infection. Viral isolation would require cell cultures and this is not what was done for diagnosis. Please revise all instances of the use of the term “isolated” in the text.

-          English language should be thoroughly revised to address major language errors.

-          Statistical reporting guidelines should be consulted. It is incorrect to report p value as = 0.000

-          Italicization for Mycoplasma pneumoniae is needed.

ABSTRACT

-          “Before the COVID-19 pandemic, respiratory pathogens such as influenza, parainfluenza, and respiratory syncytial virus were the common isolated viruses among patients with respiratory tract infections”. This phrase is incorrect, as the common pathogens are generally those related to mild respiratory illness. The authors probably meant to say that these were the most common viruses among hospitalized patients? Or among patients with severe respiratory infection?

-          Methods: Please specify whether inpatients/outpatients or both.

-          Methods should state what tests were used for diagnosis before and after the pandemic

-          Results should be completely revised, to provide the frequencies of the most important viruses before and after the pandemic, and to provide exact p value for each comparison.

-          Conclusions: “may have resulted in reduction in the rates of isolating several viruses” – this is incorrectly phrased. The authors are probably trying to explain presentation bias.

INTRODUCTION:

-          “The swift and widespread implementation of these measures was an extraordinary public health effort, aiming to curb the transmission of COVID-19 and protect vulnerable populations. Countries adopted a wide range of strategies tailored to their specific circumstances, leading to a global reduction in mobility and social interactions”. Please also address here the medical efforts made to provide care for COVID-19 patients in the beginning of the pandemic, i.e., see reference (http://www.revistamedicinamilitara.ro/wp-content/uploads/2020/04/Bucharest-mobile-military-hospital-%E2%80%93-response-to-the-COVID-19-pandemic.pdf).

METHODS:

-          Please specify whether inpatients/outpatients or both.

-          5. See comment above about the term “virus isolation”.

-          Methods should state whether the same PCR tests were routinely used for diagnosis before and after the pandemic, and with the same frequency.

-          Section 6 and in Results: mean and SD are only appropriate for parametrical variables. Was variable distribution checked?

RESULTS

-          “The mean of the included patients was 22.25 ± 28.19.” – is this age? Please specify in the text.

-          Please add measurement units in the text for all parameters: age, mmHg, %, etc. Also for all blood tests, length of hospital stay, etc. Needed for each and every variable.

-          Table 1 and 2 are redundant compared with Table 3. It would be advisable to keep only the table with the pre and post pandemic comparison. For COVID-19 vaccination which is available only in the post pandemic period, simply mention it in the text.

-          Table 3: The * should be defined in a table legend. The measurement units should be provided for each parameters. A table legend should specify the statistical tests applied in the table. The statistical reporting guidelines should be consulted. It is incorrect to report p=0.000. Is it also not enough to report only the p value. Different other statistical parameters should be reported, based on the type of test applied.

-          Figure 1 should be removed and only Fig 2 should be kept. Also, the figure should be redrawn for a more esthetic aspect.

-          Fig 2: The title mentions “viruses” but the figure also lists Mycoplasma, which is not a virus. It would be more correct to use the term “pathogens”

-          Is there data available for influenza vaccination? This would be extremely relevant

DISCUSSION

-          The Discussion should also mention what the uptake of vaccination for COVID-19 was, (see PMID 34946473) as this is very relevant to be addressed since the authors are mentioning that the pandemic has led to a more enhanced health seeking behavior among patients with comorbidities.

-          COVID-19 vaccine uptake should be discussed among study participants and with data from the general population in the country at the moment when the study was performed.

-          The same thing should be done for influenza.

-          “This is one of the first studies to assess the impact of the COVID-19 pandemic on the change in the viruses isolated in respiratory infection patients.” – this is incorrect. Many many studies have been published on this topic. For example see reference PMID 36642212

CONCLUSIONS:

-          “Although it may not be feasible to implement all the extensive interventions used during the pandemic, public health measures such as hand washing, cough etiquette, mask use and staying home during acute symptoms, may serve as useful strategies for the prevention and control of upcoming winter viruses seasons.” This is not a direct conclusion of your study and should be removed or, at best, moved to the Discussion section.

Major revision of English phrasing is needed.

Author Response

Reviewer 2:

-          The title should be revised for clearer English phrasing.

- Response: Thank you for your comment. We have revised the title for better English phrasing;

“Alterations in Patients' Clinical Outcomes and Respiratory Viral Pathogen Activity Following the COVID-19 Pandemic”

-          The term “isolated” is incorrectly used in relation to the diagnosis of viral infection. Viral isolation would require cell cultures and this is not what was done for diagnosis. Please revise all instances of the use of the term “isolated” in the text.

- Response: Thank you for your suggestion. We edited “isolated” term and replaced it with “detected”.

-          English language should be thoroughly revised to address major language errors.

- Response: Thank you for the comment. We revised the English language throughout the manuscript.

-          Statistical reporting guidelines should be consulted. It is incorrect to report p value as = 0.000

- Response: Thank you for your comment! We replaced it with P-value<0.001.

-          Italicization for Mycoplasma pneumoniae is needed.

- Response: Thank you! We edited it.

ABSTRACT

-          “Before the COVID-19 pandemic, respiratory pathogens such as influenza, parainfluenza, and respiratory syncytial virus were the common isolated viruses among patients with respiratory tract infections”. This phrase is incorrect, as the common pathogens are generally those related to mild respiratory illness. The authors probably meant to say that these were the most common viruses among hospitalized patients? Or among patients with severe respiratory infection?

- Response: Thank you for your comment! We rephrased the mentioned sentence;

“Before the COVID-19 pandemic, respiratory pathogens such as influenza, parainfluenza, and respiratory syncytial virus were the common isolated viruses among hospitalized patients with respiratory tract infections.”

-          Methods: Please specify whether inpatients/outpatients or both.

- Response: Thank you for the comment! We specified that both inpatients and outpatients were included;

“This was a retrospective observational study of in and outpatients attended Jordan University Hospital and underwent Nasopharyngeal Aspiration (NPA) in the periods from December 2017 to December 2018 and from December 2021 to December 2022.”

-          Methods should state what tests were used for diagnosis before and after the pandemic

- Response: Thank you for your comment. We added the tests used to diagnose.

“The results of multiplex respiratory pathogens real-time PCR tests for nasopharyngeal swab specimens were extracted from the electronic-based molecular diagnostic laboratory record of JUH.”

-          Results should be completely revised, to provide the frequencies of the most important viruses before and after the pandemic, and to provide exact p value for each comparison.

- Response: Thank you! We have added the frequencies and P-values for the results;

“Influenza H1N1 (8.70% vs 4.03%), influenza B (1.67% vs 0.25%), parainfluenza (1.00% vs 0.00%), human metapneumovirus (5.35% vs 0.76%), adenoviruses (6.35% vs 3.02%), and coronaviruses (8.70% vs 3.53%) were isolated in higher frequency in the period before the pandemic (P-value=0.011, 0.045, 0.045, 0.000, 0.035, 0.004).”

-          Conclusions: “may have resulted in reduction in the rates of isolating several viruses” – this is incorrectly phrased. The authors are probably trying to explain presentation bias.

- Response: Thank you for your comment. We rephrased the conclusion sentence;

“We have demonstrating reduction in the detection of several viruses which might be due to the increase of public awareness toward infection protection measures after the COVID-19 pandemic.”

INTRODUCTION:

-          “The swift and widespread implementation of these measures was an extraordinary public health effort, aiming to curb the transmission of COVID-19 and protect vulnerable populations. Countries adopted a wide range of strategies tailored to their specific circumstances, leading to a global reduction in mobility and social interactions”. Please also address here the medical efforts made to provide care for COVID-19 patients in the beginning of the pandemic, i.e., see reference (http://www.revistamedicinamilitara.ro/wp-content/uploads/2020/04/Bucharest-mobile-military-hospital-%E2%80%93-response-to-the-COVID-19-pandemic.pdf).

- Response: Thank you for your suggestion. We added the mentioned idea;

“Also, the medical efforts made to provide care for COVID-19 patients in the beginning of the pandemic burdened the resources [4].”

METHODS:

-          Please specify whether inpatients/outpatients or both.

- Response: Thank you for your comment. We addressed the mentioned issue;

“This retrospective observational study focuses on in and outpatients who sought medical attention at Jordan University Hospital (JUH) Emergency Department (ED), a tertiary adult and pediatric hospital located in Amman, Jordan.”

-          5. See comment above about the term “virus isolation”.

- Response: Thank you for your suggestion. We edited “isolated” term and replaced it with “detected”.

-          Methods should state whether the same PCR tests were routinely used for diagnosis before and after the pandemic, and with the same frequency.

- Response: Thank you for your comment. We addressed the mentioned issue.

“The same PCR tests were used in both periods with the same frequency of testing.”

-          Section 6 and in Results: mean and SD are only appropriate for parametrical variables. Was variable distribution checked?

 - Response: Thank you for the comment. The distribution was checked and it was normally distributed.

RESULTS

-          “The mean of the included patients was 22.25 ± 28.19.” – is this age? Please specify in the text.

- Response: Thank you for the comment. We edited the mentioned sentences;

“The mean age of the included patients was 22.25 ± 28.19.”

-          Please add measurement units in the text for all parameters: age, mmHg, %, etc. Also for all blood tests, length of hospital stay, etc. Needed for each and every variable.

- Response: Thank you for the comment! We added the measurements for all parameters.

-          Table 1 and 2 are redundant compared with Table 3. It would be advisable to keep only the table with the pre and post pandemic comparison. For COVID-19 vaccination which is available only in the post pandemic period, simply mention it in the text.

- Response: Thank you for the comment. The tables were edited and represented.

-          Table 3: The * should be defined in a table legend. The measurement units should be provided for each parameters. A table legend should specify the statistical tests applied in the table. The statistical reporting guidelines should be consulted. It is incorrect to report p=0.000. Is it also not enough to report only the p value. Different other statistical parameters should be reported, based on the type of test applied.

- Response: Thank you for the comment. We edited all the mentioned issues and added a table legend.

-          Figure 1 should be removed and only Fig 2 should be kept. Also, the figure should be redrawn for a more esthetic aspect.

- Response: Thank you for your comment. Both figures were represented to become clearer and more esthetic.

-          Fig 2: The title mentions “viruses” but the figure also lists Mycoplasma, which is not a virus. It would be more correct to use the term “pathogens”

- Response: Thank you for your suggestion. Mycoplasma data was removed across the whole manuscript.

-          Is there data available for influenza vaccination? This would be extremely relevant

- Response: Thank you for your comment. We totally agree with you about its relevance, however the data about influenza vaccination is not available.

DISCUSSION

-          The Discussion should also mention what the uptake of vaccination for COVID-19 was, (see PMID 34946473) as this is very relevant to be addressed since the authors are mentioning that the pandemic has led to a more enhanced health seeking behavior among patients with comorbidities.

- Response: Thank you for your suggestion. We added this idea to the discussion section;

“Another important factor is the uptake f COVID-19 vaccination which increased with time as a result of the increased awareness [28].”

-          COVID-19 vaccine uptake should be discussed among study participants and with data from the general population in the country at the moment when the study was performed.

- Response: Thank you for your suggestion. We addressed the mentioned issue in the discussion section;

“Another important factor is the uptake f COVID-19 vaccination which increased with time as a result of the increased awareness [28]. The prevalence of COVID-19 vaccination in our study was high reaching 90.7% which is similar to the rates reported in the WHO dashboard [1].”

-          The same thing should be done for influenza.

Response: Thank you for your comment. We totally agree with you about the importance of including data about influenza vaccine, however the data about influenza vaccination is not available.

-          “This is one of the first studies to assess the impact of the COVID-19 pandemic on the change in the viruses isolated in respiratory infection patients.” – this is incorrect. Many many studies have been published on this topic. For example see reference PMID 36642212

- Response: Thank you for your suggestion. We specified the sentences that it is one of the first to investigate the impact by pandemic on changes in viruses detected in respiratory infections by comparing between before and after the pandemic;

“This is one of the first studies to assess the impact of the COVID-19 pandemic on the change in the viruses detected in respiratory infection patients by comparing the detection rates between before and after the pandemic. Few studies were conducted to investigate the change in the viruses epidemiology between before and during the COVID-19 pandemic [36-38]. The majority of these studies were done in Europe, China and America with very few studies conducted in the Middle East region [36-38]. Our study adds to the literature by being conducted in a region with scarcity of data. In addition, it is one of few studies that compared the changes in viruses epidemiology between before and after the COVID-19 pandemic and not between before and during the pandemic.”

CONCLUSIONS:

-          “Although it may not be feasible to implement all the extensive interventions used during the pandemic, public health measures such as hand washing, cough etiquette, mask use and staying home during acute symptoms, may serve as useful strategies for the prevention and control of upcoming winter viruses seasons.” This is not a direct conclusion of your study and should be removed or, at best, moved to the Discussion section.

- Response: Thank you for your suggestion. We removed the mentioned sentence;

“This is one of the first studies to assess the impact of the COVID-19 pandemic on the change in the viruses detected in respiratory infection patients by comparing the detection rates between before and after the pandemic. Few studies were conducted to investigate the change in the viruses epidemiology between before and during the COVID-19 pandemic [36-38]. The majority of these studies were done in Europe, China and America with very few studies conducted in the Middle East region [36-38]. Our study adds to the literature by being conducted in a region with scarcity of data. In addition, it is one of few studies that compared the changes in viruses epidemiology between before and after the COVID-19 pandemic and not between before and during the pandemic.”

Reviewer 3 Report

Dear authors, I have read your work. The general impression is not encouraging. Although the topic is interesting and worthy of attention, there are numerous flaws that make the article unacceptable in the current version. In few words: the introduction is redundant, the methods are insufficient and unclear, the data are presented incorrectly and unclear, the discussion starts from probably incorrect premises. Some among these flaws cannot be overcome, even after an extensive revision.

In details:

-          The first thing that is unclear is the population selection. There are no clear eligibility criteria: your study population is represented by those who have presented to the Emergency Department of JUH and have performed a search for respiratory pathogens using multiplex PCR on nasotracheal aspirate. Are there any age restrictions? Is this an adult hospital or are adult and pediatric cases presented together? Is it possible to have a minimal clinical definition? Is it presumable that all the patients presented with a Febrile Respiratory Illness or is it possible that the test was also carried out for other reasons (for example for screening in subjects with a sepsis)? Have patients come by their own to the Emergency Department or are they also transferred from other hospitals? If yes, how many referred by their own and how many transferred from other hospital?

-          Did you exclude COVID-19 among these patients? With which methods?

-          I’m asking these things because your population is very strange. First of all, you say, both in results and in table 1, that the mean age is 22 years-old. Are you sure? The comorbidity profile (hypertension 50%, diabetes, respiratory and cardiovascular diseases present in a significant part) is not compatible with this age profile. It is possible that the age mean is so low because you mixed together paediatric and adult cases? If so, it is a great flaw. Adults and children should be presented separately!

-          The period of study is wrongly reported, I suppose. You wrote that the second period is from 1st dicember 2022 to 31 dicember 2022 (1 month only!). I suppose it is from the 1 dicember 2021 to 31 dicember 2022, in order to have the same time of observation of period 1 (dic 2017 to dicember 2018). If so, you cannot say that this paper analyzes respiratory viral pathogens after the COVID, since in 2021-22 Jordan was still deeply involved in COVID-19 oubreak, with 2 peaks of cases in Dicember 2021 and February 2022. This observation is during COVID-19 pandemic, and not after;

-          Patient outcomes are very strange, too. Except from Influenza A and Mycoplasma (that is a bacteria, while you in the title refer to viral pathogens only), all other viruses cause mild, self-limiting diseases. Why your hospitalization rate is 99% and ICU admission 35%? This figure is absolutely not justifiable from the diagnoses provided. Is it possible that these diagnoses are superinfections of other much more serious pathologies (COVID-19 or bacterial pneumonia), which are the real cause of the patient's hospitalization? Please discuss on it;

-          Results are presented in a wrong way. The focus of your paper is the comparison between two periods. Therefore, each population characteristic should be presented separately for period 1 and period 2. Table 1 and 2, presenting the patients all together, does not contribute to the clarity of the paper. You can do 1 or 2 tables (Table 1 for population characteristics and table 2 for isolated viruses and outcomes with 3 colums: all patients, before pandemic and during pandemic;

-          Intro is very long and redundant. For example, the paragraphs starting with “The swift and wispread implementation…” “The COVID-19 pandemic prompted governments…” and “Given the increased awareness…” all say the same concept. Please reduce. Moreover, the paragraph starting with “the increase in public health knowledge…” is not clear to me. Please rephrase;

-          Please add reference values to all laboratory and physiological paramenters. Some are very strange for me, such as  C-reactive protein (normal value 5 or 0,5?), systolic blood pressure in table 3 (30,72?) Heart rate in Table 3 (167?), Respiratory rate in table 3 (3,35?);

-          Influenza H1N1 is an Influenza A virus. Why it is presented separately? Influenza A are other Influenza A not H1N1?;

-          In the table 3, about the topic “test result” you included both “positive” (107 vs 165) and “Yes” (192 vs 232). What does it mean?

-          Figures are not easy to read in the current form. You should couple columns for each pathogen, showing period 1 next to period 2, in order to give a easy visual interpretation, too. Don’t use too many colours, it is confounding;

-          Last but not least: are you sure that respiratory pathogens really decreased? You calculated percentage on the total of patients tested (298 vs 397, about 20% more in the second period). Given that the test accuracy is not complete, and that the interference of Covid-19 was still very present in the period 2, you can not say nothing of definitive about it. Please note that the overall number of patients tested increased, and that the difference on the overall positivity rate is not significant. The percentage of positivity for each agent is decreased, because it is calculated respect to the total of patients who performed the test, that are 20% more, consequently a decrease for each agent is expected. If you repeat the statistical analysis comparing rates of positivity for each agent using as total the number of total positive tests, you will probably find no statistical differences.

Few improvements are necessary, but in general it is well written 

Author Response

Reviewer 3:

Dear authors, I have read your work. The general impression is not encouraging. Although the topic is interesting and worthy of attention, there are numerous flaws that make the article unacceptable in the current version. In few words: the introduction is redundant, the methods are insufficient and unclear, the data are presented incorrectly and unclear, the discussion starts from probably incorrect premises. Some among these flaws cannot be overcome, even after an extensive revision.

- Response: Thank you so much for your efforts and time in reviewing our manuscript! We complied with the comments you provided.

In details:

-          The first thing that is unclear is the population selection. There are no clear eligibility criteria: your study population is represented by those who have presented to the Emergency Department of JUH and have performed a search for respiratory pathogens using multiplex PCR on nasotracheal aspirate. Are there any age restrictions? Is this an adult hospital or are adult and pediatric cases presented together? Is it possible to have a minimal clinical definition? Is it presumable that all the patients presented with a Febrile Respiratory Illness or is it possible that the test was also carried out for other reasons (for example for screening in subjects with a sepsis)? Have patients come by their own to the Emergency Department or are they also transferred from other hospitals? If yes, how many referred by their own and how many transferred from other hospital?

- Response: Thank you so much for these comments. We edited the methods section to comply with your suggestions.

“Methods:

This retrospective observational study focuses on patients who sought medical attention at Jordan University Hospital (JUH) Emergency Department (ED), a tertiary adult and pediatric hospital located in Amman, Jordan. Patients from all age groups who underwent Nasopharyngeal Aspiration (NPA) with viral Polymerase Chain Reaction (PCR) were included in the study, covering the periods from 1st of May 2017 to 30th of June 2018 and from 1st of May 2022 to 30th of June 2023. NPA with PCR for pathogen detection was performed on patients who presented by their own (not transferred from another hospital) with symptoms of respiratory viral infection (febrile respiratory illness), based on the ED physician's decision.”

-          Did you exclude COVID-19 among these patients? With which methods?

- Response: Thank you for your comment. Yes, COVID-19 patients were excluded and any patient their nasal swab result was positive for COVID-19 was excluded from the analysis;

“Methods

Patients were excluded from the study if they had COVID-19 infection (tested by PCR nasal swab), immunosuppression, including transplant patients, patients with cancer, and those receiving immunosuppressive drugs or chemotherapy.”

-          I’m asking these things because your population is very strange. First of all, you say, both in results and in table 1, that the mean age is 22 years-old. Are you sure? The comorbidity profile (hypertension 50%, diabetes, respiratory and cardiovascular diseases present in a significant part) is not compatible with this age profile. It is possible that the age mean is so low because you mixed together paediatric and adult cases? If so, it is a great flaw. Adults and children should be presented separately!

- Response: Thank you for drawing our attention to this issue! There was a typo in Table 1 which we fixed. The prevalence of comorbidities (diabetes, hypertension, and cardiovascular diseases) in the population currently ranged between 14.1% to 17.7%. We were not able to present the data of paediatrics and adults separately due to the low sample size;

Table 1. The General Demographics of the Participants.

Variable

Response

Frequency

Percentage (%)

Sex

Male

362

52.1

Female

333

47.9

Hypertension

No

570

81.9

Yes

123

17.7

Diabetes Mellitus

No

570

82.3

Yes

123

17.7

Respiratory Diseases

No

506

72.7

Yes

190

27.3

Cardiovascular Diseases

No

598

85.9

Yes

98

14.1

Variable

Mean

SD

Range

Age (years)

22.25

28.19

0.01-96

Systolic Blood Pressure

135.56

69.7

59-186

Diastolic Blood Pressure

65.41

12.81

30-110

Heart Rate

82.24

56.85

35-150

Respiratory Rate

27.81

11.13

5-95

Temperature

37.03

2.50

36.0-40.0

O2 Saturation

92.21

6.66

58-100

-          The period of study is wrongly reported, I suppose. You wrote that the second period is from 1st dicember 2022 to 31 dicember 2022 (1 month only!). I suppose it is from the 1 dicember 2021 to 31 dicember 2022, in order to have the same time of observation of period 1 (dic 2017 to dicember 2018). If so, you cannot say that this paper analyzes respiratory viral pathogens after the COVID, since in 2021-22 Jordan was still deeply involved in COVID-19 oubreak, with 2 peaks of cases in Dicember 2021 and February 2022. This observation is during COVID-19 pandemic, and not after;

- Response: Thank you so much for drawing our attention to this issue! We totally agree with you. There was a mistake in the mentioned sentence and we edited it. We are totally aware of it now. The chosen period was from 1st of May 2022 to 30th of June 2023 which corresponds to the period after the pandemic as Jordan did not have any surge in COVID-19 cases during that period. We have also provided the raw data as a supplementary material which presents the exact dates for each test included in the analysis.

“The period before the pandemic was considered from 1st of May 2017 to 30th of June 2018 while the period after it was from 1st of May 2022 to 30th of June 2023.”

-          Patient outcomes are very strange, too. Except from Influenza A and Mycoplasma (that is a bacteria, while you in the title refer to viral pathogens only), all other viruses cause mild, self-limiting diseases. Why your hospitalization rate is 99% and ICU admission 35%? This figure is absolutely not justifiable from the diagnoses provided. Is it possible that these diagnoses are superinfections of other much more serious pathologies (COVID-19 or bacterial pneumonia), which are the real cause of the patient's hospitalization? Please discuss on it;

- Response: Thank you for your comment! The hospitalization rate was high cause this is a cohort study from a tertiary centre so the data might be shifted to the most severe cases. The patients’ presentation was with respiratory febrile illness and not any other more severe presentations. In addition, COVID-19 patients were excluded from the analysis. We added this limitation to the limitations section. Regarding including mycoplasma, we totally agree with you and we have removed mycoplasma data from the manuscript;

“Moreover, this study was conducted in a tertiary hospital, thus our results might be shifted to patients with more severe infections which is indicated by the high hospitalization rate.”

-          Results are presented in a wrong way. The focus of your paper is the comparison between two periods. Therefore, each population characteristic should be presented separately for period 1 and period 2. Table 1 and 2, presenting the patients all together, does not contribute to the clarity of the paper. You can do 1 or 2 tables (Table 1 for population characteristics and table 2 for isolated viruses and outcomes with 3 colums: all patients, before pandemic and during pandemic;

- Response: Thank you for your suggestion! We edit the tables to represent the data. Table 1 for the general characteristics of the population, Table 2 for the COVID-19 infection and vaccination history, Table 3 for the Patients Outcomes, Laboratory Investigations and Characteristics of Patients Admitted to the ICU, and Table 4 for Differences in the Demographics between Patients before and after the Pandemic.

-          Intro is very long and redundant. For example, the paragraphs starting with “The swift and wispread implementation…” “The COVID-19 pandemic prompted governments…” and “Given the increased awareness…” all say the same concept. Please reduce. Moreover, the paragraph starting with “the increase in public health knowledge…” is not clear to me. Please rephrase;

- Response: Thank you so much for your comment. We totally agree with you. We reduced the redundancy and paraphrased the Introduction section.

-          Please add reference values to all laboratory and physiological paramenters. Some are very strange for me, such as  C-reactive protein (normal value 5 or 0,5?), systolic blood pressure in table 3 (30,72?) Heart rate in Table 3 (167?), Respiratory rate in table 3 (3,35?);

- Response: Thank you for your suggestions. We added the physiological values for all parameters. In addition, we fixed the mentioned typos in table 3.

-          Influenza H1N1 is an Influenza A virus. Why it is presented separately? Influenza A are other Influenza A not H1N1?;

- Response: Thank you so much for your comment. Influenza A are other Influenza A not H1N. We added a clarification on that in the figures footnote;

Figure 1. Distribution of Viruses in Patients’ Samples.

Influenza A: Influenza Type A other than H1N1, Influenza H1N1: Influenza subtype hemagglutinin1 neuraminidase 1, Influenza B: Influenza Type B, RSV: Respiratory Syncytial Virus, HMPV: Human Metapneumovirus.

Figure 2. Distribution of Isolated Viruses between Before and After the Pandemic.

Influenza A: Influenza Type A other than H1N1, Influenza H1N1: Influenza subtype hemagglutinin1 neuraminidase 1, Influenza B: Influenza Type B, RSV: Respiratory Syncytial Virus, HMPV: Human Metapneumovirus.

-          In the table 3, about the topic “test result” you included both “positive” (107 vs 165) and “Yes” (192 vs 232). What does it mean?

- Response: Thank you for your comment. It is the percentage of positive and negative tests. We edited the mentioned issue in table 3.

-          Figures are not easy to read in the current form. You should couple columns for each pathogen, showing period 1 next to period 2, in order to give a easy visual interpretation, too. Don’t use too many colours, it is confounding;

- Response: Thank you for your suggestion. We represented figure 2 again and reduced the number of colours.

Figure 2. Distribution of Isolated Viruses between Before and After the Pandemic.

Influenza A: Influenza Type A other than H1N1, Influenza H1N1: Influenza subtype hemagglutinin1 neuraminidase 1, Influenza B: Influenza Type B, RSV: Respiratory Syncytial Virus, HMPV: Human Metapneumovirus.

-          Last but not least: are you sure that respiratory pathogens really decreased? You calculated percentage on the total of patients tested (298 vs 397, about 20% more in the second period). Given that the test accuracy is not complete, and that the interference of Covid-19 was still very present in the period 2, you can not say nothing of definitive about it. Please note that the overall number of patients tested increased, and that the difference on the overall positivity rate is not significant. The percentage of positivity for each agent is decreased, because it is calculated respect to the total of patients who performed the test, that are 20% more, consequently a decrease for each agent is expected. If you repeat the statistical analysis comparing rates of positivity for each agent using as total the number of total positive tests, you will probably find no statistical differences.

- Response: Thank you so much for your comment. The COVID-19 surge was already subsided in the period after the pandemic which limits its effect on our results. The difference in the total number and positive tests was not significant indicating that calculating the percentage of viruses according to the grand total or positive test will not impact the results. On the other hand, calculating it without taking into consideration the percentage of negative tests might result in ignoring a part of the study population who had negative results which might result in reducing the power of the tests and increasing the possibility of type 1 and 2 errors.

Reviewer 4 Report

Thank you for the opportunity to review this interesting article. It is a retrospective observational study of patients attended Jordan University Hospital and underwent Nasopharyngeal Aspiration (NPA) in the periods from December 2017 to December 2018 and from December 2021 to December 2022. The authors aimed to compare the prevalence of the isolated viruses as well as the patients’ characteristics and outcomes. The content of the study is simple, and the analysis is basic.

There are several concerns that the authors should address in current manuscript:

(1).    There are multiple studies that have analyzed this topic (PMID: 36721137, PMID: 35350252, PMID: 34181015, PMID: 36628396, and etc). What does your manuscript add?

(2).    Materials and methods: This study has been approved by the ethics committee of our hospital, and written informed consent has been obtained from all patients. Please provide the IRB approval number in your article.

(3).    Figure 1 and 2: Abbreviations should be defined.

Author Response

Reviewer 4:

Thank you for the opportunity to review this interesting article. It is a retrospective observational study of patients attended Jordan University Hospital and underwent Nasopharyngeal Aspiration (NPA) in the periods from December 2017 to December 2018 and from December 2021 to December 2022. The authors aimed to compare the prevalence of the isolated viruses as well as the patients’ characteristics and outcomes. The content of the study is simple, and the analysis is basic.

There are several concerns that the authors should address in current manuscript:

- Response: Thank you so much for reviewing our manuscript! We appreciate your time and efforts! We complied with the comments mentioned below.

(1).    There are multiple studies that have analyzed this topic (PMID: 36721137, PMID: 35350252, PMID: 34181015, PMID: 36628396, and etc). What does your manuscript add?

- Response: Thank you for your comment. We have added the following paragraph in the discussion section to describe what our study adds to the literature;

“This is one of the first studies to assess the impact of the COVID-19 pandemic on the change in the viruses isolated in respiratory infection patients. Few studies were conducted to investigate the change in the viruses epidemiology between before and during the COVID-19 pandemic [34-36]. The majority of these studies were done in Europe, China and America with very few studies conducted in the Middle East region [34-36]. Our study adds to the literature by being conducted in a region with scarcity of data. In addition, it is one of few studies that compared the changes in viruses epidemiology between before and after the COVID-19 pandemic and not between before and during the pandemic.”

(2).    Materials and methods: This study has been approved by the ethics committee of our hospital, and written informed consent has been obtained from all patients. Please provide the IRB approval number in your article.

- Response: Thank you for your comment. We have added the IRB approval number;

“This study adhered to human ethical practices and the principles outlined in the Declaration of Helsinki. The Institutional Review Board (IRB#223000353) at the University of Jordan approved the study, waiving the need for informed consent.”

(3).    Figure 1 and 2: Abbreviations should be defined.

- Response: Thank you for your suggestion. We have added the abbreviations to figure 1 and 2;

Figure 1. Distribution of Viruses in Patients’ Samples.

Influenza A: Influenza Type A, Influenza H1N1: Influenza subtype hemagglutinin1 neuraminidase 1, Influenza B: Influenza Type B, RSV: Respiratory Syncytial Virus, HMPV: Human Metapneumovirus.

Figure 2. Distribution of Isolated Viruses between Before and After the Pandemic.

Influenza A: Influenza Type A, Influenza H1N1: Influenza subtype hemagglutinin1 neuraminidase 1, Influenza B: Influenza Type B, RSV: Respiratory Syncytial Virus, HMPV: Human Metapneumovirus.

Round 2

Reviewer 2 Report

I thank the authors for addressing my previous comments

Language revision is still needed.

Author Response

Thank you so much for your efforts and time in improving the article! We really appreciate it!

Reviewer 3 Report

I understand that authors tried to improve the paper, but I repeat, as already said after first revision, that some flaws cannot be addressed. Despite the paper is improved, is is still largely inadequate for publication.

- you can not present togheter adult and paediatric population. Adults and paediatric populations are two different population that must be presented separately. It changes the epidemiology, the clinical presentation and outcomes, the managment, everything is different among a <1 year baby and an adult. Presenting togheter the two population (including many issues that are presented as general mean) is a non-sense;

- I remain with my opinion that the clinical presentation and outcomes af these patients is too strange, and need a justification. You said that mostly were oupatients, and that more "at-risk" categories, such as immunosuppressed, were excluded (in the meanwhile, how do you define immunosuppression?). Neverthless, out of a population of 695 outpatients with mild, self-limiting diseases such as Influenza, parainfluenza, RSV, adenovirus, 99% is admitted in hospital! It is simply not possible, some relevant bias must be present. Similarly, it is not possible that these patients have a mean PCR of 50. This is a value compatible with severe bacterial infection,and not with mild viral diseases. In few words, your population is clinically too severe to be affected by the reported diseases. There is something of wrong in your data;

- most of my suggestions were not addressed. Table 1 and 2 still present data all togheter,and not divided before and after the pandemic, the figure is still difficult to be read, and my suggestion was not addressed;

- most of mistakes were not corrected. For example, in table 3, systolic brood pressure is still 30 and 15 (?), heart rate is still 112 and 167 (??), and restiratory rat is still 3 (???);

- finally, I have still many doubts about the results. Look at the table 3: test results (that are still positive and yes where shoud be positive and negative) show that before pandemic 299 patients were tested with 192 positive test (35,8%), after the pandemic 397 patients were tested (about 100 patients more) with 165 positive results (41,6). Thus, the overal positivity increased, even if not significantly. The total of viruses identified is more that the total reported: 241 viruses were reported before pandemic (instead of 192), and 230 after the pandemic (instead of 165). Are there patients with more than one virus isolated? How many are the coinfections? How many before and after the pandemic? Are the coinfections more frequent among adults or children? How the coinfections impact on the statistical analysis? Are you comparing changes in virus or changes in number of patients? And more, how about the accuracy of test? Increasing the number of patients tested, you increase two different phenomena: the number of false negative test, and the numeber of patients with more that one isolates. In both cases, if you analyse the single virus, they appear to be reducing, but it is not a true finding, but only a statistical changes due different bias. Looking critically at your results, the rates of isolation of respiratory viruses before and after the pandemic are probably the same.

English is good

Author Response

Response Letter:

I understand that authors tried to improve the paper, but I repeat, as already said after first revision, that some flaws cannot be addressed. Despite the paper is improved, is is still largely inadequate for publication.

- Response: Thank you so much for your efforts and time in reviewing the manuscript! We complied and replied to your comments mentioned below.

- you can not present togheter adult and paediatric population. Adults and paediatric populations are two different population that must be presented separately. It changes the epidemiology, the clinical presentation and outcomes, the managment, everything is different among a <1 year baby and an adult. Presenting togheter the two population (including many issues that are presented as general mean) is a non-sense;

- Response: Thank you so much for your suggestion. We represented the data in Table 1 and 3 according to age (adults and pediatrics);

Results;

“The total number of the included patients was 695 of them, 54.4% of them were males among the pediatrics while 48.0% were males among adults. The prevalence of hypertension and diabetes were 11.2% and 41.5% among adults whereas they were 1.1% and 2.0% among pediatrics, respectively.. The percentage of respiratory and cardiovascular diseases were 30.9% and 29.3% among adults and 25.3% and 5.6% among pediatrics, respectively. The mean systolic and diastolic blood pressure were 122.69 ± 21.41mmHgand 72.33 ± 13.07mmHg among adults while the means were 142.73 ± 81.7mmHg and 61.54 ± 10.89 among pediatrics, respectively. Among adults, the mean body temperature was 37.12 ± 0.82c and the mean O2 saturation was 91.45 ± 7.61%. The mean body temperature and O2 saturation among pediatrics were 36.98 ± 3.05 and 92.63 ± 6.04, respectively. Table 1 shows the characteristics of the included patients while table 2 demonstrates the characteristics of COVID-19 infection and vaccination history.

The majority of the patients were hospitalized (98.4%) and 23.0% of the patients were admitted to the ICU among adults. Similarly, 99.6% were hospitalized and 36.1% were admitted to the ICU among pediatrics. The prevalence of hypertension and diabetes among adults admitted to the ICU was 47.4% and 35.1%, respectively while they were 3.1% and 5.6% among pediatrics. Moreover, 60.9% of the adult patients needed O2 devices and 14.2% of them needed intubation. Among adults, the mean CRP was 85.76 ± 97.75 and the mean WBC count was 11.55 ± 9.58. While, among pediatrics, the mean CRP and WBC were 33.79 ± 54.85 and 12.47 ± 9.79, respectively. In addition, the mean length of the hospital stay was 11.45 ± 11.02 among adults while it was 12.47 ± 9.79. Moreover, 13.8% and 3.4% died among adult and pediatric patients, respectively. Table 3 displays the laboratory investigation results, patients’ outcomes and characteristics of patients admitted to the ICU.”

- I remain with my opinion that the clinical presentation and outcomes af these patients is too strange, and need a justification. You said that mostly were oupatients, and that more "at-risk" categories, such as immunosuppressed, were excluded (in the meanwhile, how do you define immunosuppression?). Neverthless, out of a population of 695 outpatients with mild, self-limiting diseases such as Influenza, parainfluenza, RSV, adenovirus, 99% is admitted in hospital! It is simply not possible, some relevant bias must be present. Similarly, it is not possible that these patients have a mean PCR of 50. This is a value compatible with severe bacterial infection,and not with mild viral diseases. In few words, your population is clinically too severe to be affected by the reported diseases. There is something of wrong in your data;

- Response: Thank you for your comment. We reviewed our data again. All the tests were done in the ER department. However, this might be explained by two issues; the first is that this study was done in a tertiary referral hospital which might shift our data to the severe cases while the second is the high cost and low availability of such a test so it is mainly done when the attending physician suspect severe diseases. Yet, it is important to note that both issues were before and after the pandemic so it is not expected to affect our results. We added these points to the limitations section;

Moreover, this study was conducted in a tertiary referral hospital while the PCR test is considered of high cost and low availably. Thus, our results might be shifted to patients with more severe infections which is indicated by the high hospitalization rate. Yet, it is important to note that these limitations were in the before and after pandemic periods, hence it is less likely to affect our results.”

- most of my suggestions were not addressed. Table 1 and 2 still present data all togheter,and not divided before and after the pandemic, the figure is still difficult to be read, and my suggestion was not addressed.

- Response: Thank you for your comment. We separated the data in table 1 and 3 according to age while we represented the data in table 4 divided to before and after the pandemic.

- most of mistakes were not corrected. For example, in table 3, systolic brood pressure is still 30 and 15 (?), heart rate is still 112 and 167 (??), and restiratory rat is still 3 (???);

- Response: Thank you for your comment. We edited the typos in the mentioned numbers.

- finally, I have still many doubts about the results. Look at the table 3: test results (that are still positive and yes where shoud be positive and negative) show that before pandemic 299 patients were tested with 192 positive test (35,8%), after the pandemic 397 patients were tested (about 100 patients more) with 165 positive results (41,6). Thus, the overal positivity increased, even if not significantly. The total of viruses identified is more that the total reported: 241 viruses were reported before pandemic (instead of 192), and 230 after the pandemic (instead of 165). Are there patients with more than one virus isolated? How many are the coinfections? How many before and after the pandemic? Are the coinfections more frequent among adults or children? How the coinfections impact on the statistical analysis? Are you comparing changes in virus or changes in number of patients? And more, how about the accuracy of test? Increasing the number of patients tested, you increase two different phenomena: the number of false negative test, and the numeber of patients with more that one isolates. In both cases, if you analyse the single virus, they appear to be reducing, but it is not a true finding, but only a statistical changes due different bias. Looking critically at your results, the rates of isolation of respiratory viruses before and after the pandemic are probably the same.

- Response: Thank you so much for your drawing our attention to this issue. We reanalyzed the data after matching the patients in the number of patients before and after the pandemic randomly and similar results to the primary analysis was noted. We added the percentage of coinfection before and after the pandemic which were similar before and after the pandemic. The matching in the number of tested patients add to the evidence of our main analysis as it should resolve any potential bias arising from the accuracy of the tests, percentage of positive patients and number of tested patients;

Methods

“In addition, we carried out a random matching process between the two periods to equalize the number of tests between the two groups. A random sample, that equals the before pandemic group in number, was taken from after the pandemic group. This was followed by assessing the differences between the two groups in the viruses distribution using chi-square.”

Results

“After matching the number of tested patients before and after the pandemic, similar results to the primary analysis were found (Table 4).”

Variable

Before Pandemic

(n=298)

After Pandemic

(n=397)

P-value

Gender

Male

147

(49.3)

215

(54.2)

0.207

Female

151

(50.7)

182

(45.8)

Age

55.13 ± 21.52

59.08 ± 19.06

0.353

Respiratory Diseases

No

220

(73.6)

286

(72.0)

0.652

Yes

79

(26.4)

111

(28.0)

Hypertension

No

224

(75.4)

346

(87.4)

0.000*

Yes

73

(24.6)

50

(12.6)

Diabetes

No

233

(78.7)

348

(87.9)

0.001*

Yes

63

(21.3)

48

(12.1)

Cardiovascular Diseases

No

253

(84.6)

345

(86.9)

0.391

Yes

46

(46.9)

52

(53.1)

Test Results

No

107

(35.8)

165

(41.6)

0.122

Yes

192

(64.2)

232

(58.4)

Coinfection

No

250

(83.6)

332

(83.7)

0.900

Yes

49

(16.4)

65

(16.3)

Viruses (Before matching the sample size of the two groups)

Influenza A

29

(9.7)

40

(10.1)

0.869

Influenza H1N1

26

(8.7)

16

(4.0)

0.011*

Influenza B

5

(1.7)

1

(0.3)

0.045*

Coronavirus

26

(8.7)

14

(3.5)

0.004*

Parainfluenza

3

(1.0)

0

(0.0)

0.045*

RSV

54

(18.1)

63

(15.9)

0.444

HMPV

16

(5.2)

3

(0.8)

0.000*

Adenovirus

19

(6.4)

12

(3.0)

0.035*

Rhinovirus/Enterovirus

63

(21.1)

81

(20.4)

0.830

Viruses (After Matching the sample size of the two groups)

Influenza A

29

(9.7)

26

(8.7)

0.681

Influenza H1N1

26

(8.7)

10

(2.98)

0.048*

Influenza B

5

(1.7)

1

(0.3)

0.045*

Coronavirus

26

(8.7)

6

(2.0)

0.000*

Parainfluenza

3

(1.0)

0

(0.0)

0.045*)

RSV

54

(18.1)

55

(50.5)

0.900

HMPV

16

(5.2)

2

(0.7)

0.001*

Adenovirus

19

(6.4)

7

(2.7)

0.049

Rhinovirus/Enterovirus

63

(21.1)

63

(21.1)

1.000

Hospitalization

No

1

(0.3)

5

(1.3)

0.191

Yes

298

(99.7)

392

(98.7)

ICU Admission

No

193

(65.0)

283

(71.3)

0.077

Yes

104

(35.0)

114

(28.7)

Use of O2 Devices

No

104

(34.9)

194

(48.9)

0.000*

Yes

194

(65.1)

203

(51.1)

Intubation

No

266

(89.0)

348

(88.3)

0.793

Yes

33

(11.0)

46

(11.7)

Death

No

276

(92.6)

369

(93.2)

0.774

Yes

22

(7.4)

27

(6.8)

Systolic Blood Pressure (normal for pediatric: 60-131, for adults: 90-120)

130.72 ± 30.59

115.88 ± 24.41

0.252

Diastolic Blood Pressure (normal for pediatric: 31-83, for adults: 60-80)

69.08 ± 12.69

62.65 ± 12.20

0.000*

Heart Rate (normal for pediatric: 60-200, for adults: 60-100)

102.20 ± 29.37

107 ± 75.27

0.385

Temperature (normal: 36.5-37.3c)

37.13 ± 2.26

36.96 ± 2.66

0.371

O2 Saturation (normal: 95%-100%)

92.07 ± 6.06

92.31 ± 7.08

0.620

Respiratory Rate (normal for pediatric: 12-60, for adults: 12-18)

23.35 ± 1.87

23.07 ± 1.94

0.320

C-reactive Protein (normal: <0.3 mg/dl)

54.85 ± 84.04

50.64 ± 71.76

0.484

White Blood Cells (normal: 4.5*10^3 to 11.0* 10^3)

10.49 ± 8.16

12.37 ± 10.48

0.008*

Neutrophils (normal: 40-60%)

57.73 ± 22.77

58.20 ± 22.11

0.787

Lymphocytes (normal: 20-40%)

31.68 ± 20.70

32.35 ± 20.25

0.670

Length of Hospital Stay

12.21 ± 20.24

9.47 ± 12.26

0.038*

Reviewer 4 Report

All the questions were adequately addressed.

Author Response

(The authors gave the same response as above.)
